# Infection and Coinfection of Porcine-Selected Viruses (PPV1 to PPV8, PCV2 to PCV4, and PRRSV) in Gilts and Their Associations with Reproductive Performance

**DOI:** 10.3390/vetsci11050185

**Published:** 2024-04-24

**Authors:** Diana S. Vargas-Bermudez, Andres Diaz, Gina Polo, Jose Dario Mogollon, Jairo Jaime

**Affiliations:** 1Universidad Nacional de Colombia, Sede Bogotá, Facultad de Medicina Veterinaria y de Zootecnia, Departamento de Salud Animal, Centro de Investigación en Infectología e Inmunología Veterinaria—CI3V.Cra. 30 # 45-03, Bogotá 11001, Colombia; dsvargasb@unal.edu.co (D.S.V.-B.); josedmogollon@yahoo.es (J.D.M.); 2Pig Improvement Company, Hendersonville, TN 37075, USA; andres.diaz@genusplc.com; 3Instituto de Salud Pública, Pontificia Universidad Javeriana, Bogota 110231, Colombia; pologp@javeriana.edu.co

**Keywords:** porcine parvoviruses (PPVs), novel porcine parvoviruses (nPPVs), PCV2, PCV3, PRRSV, porcine reproductive failure (PRF), gilts, coinfections

## Abstract

**Simple Summary:**

Different viral pathogens, some considered primary agents, affect the reproductive performance of gilts and sows, such as circoviruses 2 and 3, parvovirus 1, and the PRRS virus. Recently, new viruses have been discovered, such as novel parvoviruses (named 2 to 8), whose effects on reproductive performance are not evident. In this study, we evaluate the presence of these new parvoviruses together with the primary viruses in a particular group: the gilts. This group is essential since they are the ones that will replace the sows that leave the reproductive cycle in the herd. We found the presence of these new viruses (except parvovirus 8) simultaneously in the same gilt with primary viruses (in addition to sextuple coinfections). Furthermore, we found that some primary viruses may be associated with the presence of some parvoviruses, such as 6 and 5. Likewise, parvoviruses 4 and 6 could affect important reproductive variables, such as the farrowing rate. This observational field study attempts to draw attention to and be a starting point in the search for answers about the effect of these selected viruses in the group of gilts and their impact on reproductive performance.

**Abstract:**

Seven novel porcine parvoviruses (nPPVs) (PPV2 through PPV8) have been described, although their pathogenicity and possible effects on porcine reproductive failure (PRF) are undefined. In this study, these nPPVs were assessed in gilts from Colombia; their coinfections with PPV1, PCV2, PCV3, PCV4, and PRRSV and an association between the nPPVs and the reproductive performance parameters (RPPs) in sows were determined. For this, 234 serum samples were collected from healthy gilts from 40 herds in five Colombian regions, and the viruses were detected via real-time PCR. The results confirmed the circulation of PPV2 through PPV7 in Colombia, with PPV3 (40%), PPV5 (20%), and PPV6 (17%) being the most frequent. Additionally, no PCV4 or PPV8 was detected. PPV2 to PPV7 were detected in concurrence with each other and with the primary PRF viruses, and these coinfections varied from double to sextuple coinfections. Additionally, the association between nPPVs and PRF primary viruses was statistically significant for the presence of PPV6 in PCV3-positive (*p* < 0.01) and PPV5 in PPRSV-positive (*p* < 0.05) gilts; conversely, there was a significant presence of PPV3 in both PCV2-negative (*p* < 0.01) and PRRSV-negative (*p* < 0.05) gilts. Regarding the RPPs, the crude association between virus detection (positive or negative) and a high or low RPP was only statistically significant for PCV3 and the farrowing rate (FR), indicating that the crude odds of a low FR were 94% lower in herds with PCV3-positive gilts. This finding means that the detection of PCV3 in gilts (PCV3-positive by PCR) is associated with a higher FR in the farm or that these farms (with positive gilts) have lower odds (OR 0.06, *p*-value 0.0043) of a low FR. Additionally, a low FR tended to be associated with the detection of PPV4 and PPV5 (*p*-value < 0.20). This study is important for establishing the possible participation of nPPVs in PRF.

## 1. Introduction

Parvoviruses (PVs) are non-enveloped viruses with a linear ssDNA genome and are classified within the *Parvoviridae* family that infects invertebrates and vertebrates [1]. In recent years, with the advances in high-throughput sequencing techniques and metagenomics, novel PVs have been discovered in different animal species [2]. In swine, eight porcine parvoviruses (PPVs) have been described (PPV1 through PPV8) [3,4,5]. PPV1 is endemic in swine populations and is known to have been circulating in pigs since the 1960s. It is associated with porcine reproductive failure (PRF), being the primary agent responsible for SMEDI syndrome that causes stillbirths, mummies, embryonic mortality, and sow infertility [6]. Although the DNA sequence of the novel PPV (nPPV) species (PPV2 through PPV8) has already been determined, their pathogenic and clinical manifestations are still unknown [2]. Different studies have shown a wide geographical distribution of these nPPVs [7]. PPV2 was first reported in 2001 in Myanmar from serum samples [3] and has since been reported in different regions worldwide with variable prevalence rates and in various types of samples [8,9,10,11,12,13,14,15]. Recently, the detection of PPV2 by in situ hybridization (ISH) in alveolar macrophages of lungs with interstitial pneumonia, together with high viral loads in the same tissue, suggests that PPV2 is part of the porcine respiratory disease complex (PRDC) [16]. In the case of PPV3, it was reported in 2008 from tissue samples collected at slaughter plants in Hong Kong [17] and subsequently in different countries with variable prevalences and in various types of samples [8,13,18]. Furthermore, PPV3 was identified as highly prevalent in wild boars from Germany [19,20]. For PPV4, it was described simultaneously in 2010 in the USA in pigs with porcine circovirus-associated diseases (PCVADs) [21] and in China in pigs with PRDC and aborted fetuses [22]. PPV4 has been found less frequently than the other PPVs; however, its presence in reproductive samples such as semen and abortions [8], as well as ovaries and uteruses [23], suggests that it could be associated with porcine reproductive failure (PRF). In regard to PPV5, it was found in 2013 in the lungs of pigs with different syndromes [13,24], while in 2014, PPV6 was detected in China in samples from abortions of breeding sows, piglets, and finishing pigs [25]. Both PPV5 and PPV6 have been reported in countries such as the USA, Mexico, China, Korea, Poland, and Brazil and likewise in different samples such as serum, oral fluids, lungs, stools, chops, and fetal hearts [9,15,26,27,28]. Moving to PPV7, it was also described in 2016 in the USA in serum samples, stools, and nasal swabs. Subsequently, it was found in Korea [29], China [30], some European countries [31], and Colombia [32]. Finally, for PPV8, it was described for the first time in September 2022 in China from lung samples [5].

The causal relationship of these nPPVs with different diseases or clinical syndromes has been challenging to ascertain because Koch’s postulates remain to be fulfilled. PPVs have been reported in different countries, in different types of samples, and also in healthy pigs. Another relevant aspect of PPVs is that they have been found in concurrent infections in domestic swine and wild boars with other viral pathogens, such as porcine circovirus type 2 (PCV2), porcine circovirus type 3 (PCV3), and porcine reproductive and respiratory syndrome virus (PRRSV) [7,13,14,15,33,34]. The clinical pathologic effects of these coinfections in the field are unknown. In Colombia, it is known that PCV2 and PRRSV circulate endemically and cause epidemic outbreaks in swine herds. PCV2 is controlled by vaccination with immunization protocols of piglets at three weeks of age and gilts; some herds also vaccinate sows [35]. On the other hand, PRRSV has been circulating in Colombia since 1996, and sporadic outbreaks are controlled by gilt acclimation (exposure to feedback) and herd closure [36,37]. Despite the existence of PRRSV-licensed vaccines, none of them are used to control PRRSV infections in Colombia. Finally, PCV3 has been described in Colombia since 2019 as being associated with PRF [38,39]. The simultaneous presence of PPV1, PCV2, PRRSV, and PCV3, as well as the nPPV species, may be involved in PRF. Hence, the objectives of this study were (i) to determine the presence via real-time PCR of the nPPV species (PPV2 through PPV8) in Colombia; (ii) to establish their interactions, in terms of coinfection, with other viral agents (PPV1, PCV2, PCV3, PCV4, and PRRSV); and (iii) to explore the possible associations of these nPPVs and reproductive performance parameters (RPPs) in sow herds. This study was conducted using gilts (180–200 days of age) as a target group, considering that the health status of this productive group is of great relevance in swine production because reproductive failures in these animals affect the performance of the sow herd.

## 2. Materials and Methods

### 2.1. Sample Selection and Reproductive Performance Parameters

Sera from 234 gilts (180–200 days of age) were collected within 40 herds in Colombia’s five leading swine production regions. Farms with >70 gilts (based on the sow’s population census) (https://www.ica.gov.co/areas/pecuaria/servicios/epidemiologia-veterinaria/censos-2016/censo-2018.aspx (accessed on 1 June 2020)) were defined as the primary sampling units and then divided into five strata according to the spatial location (region). After the random selection of the farm, gilts within each farm were considered the secondary sampling unit. Eight herds per region were randomly selected, and 5 to 6 serum samples were conveniently collected per herd. The sample size calculation of farms and gilts was based on the number of farms with >70 gilts in each region. The five regions selected and the total number of samples per region were Atlántico (*n* = 45), Antioquia (*n* = 47), Cundinamarca (*n* = 41), Eje Cafetero (*n* = 52), and Valle del Cauca (*n* = 49). The number of gilts to be examined was determined as six per herd to estimate the presence/absence of the viruses, assuming 66% prevalence between herds, 40% prevalence within herds, a 96% test sensitivity, a 99% test specificity, and a 5% error. Sample size calculations were performed considering a hypergeometric distribution using the “epiR” package from the computational language for statistical analysis R [40]. Management of gilts was analyzed based on herd data, such as source, acclimatization practices (e.g., exposure of gilts to oral fluids, feces, or mummies), vaccination scheme, replacement rate, and herd size; these were recovered in a survey (Table 1). Additionally, it registered key RPPs such as the farrowing rate (FR), percentage of stillbirths (SBs), percentage of mummies (MMs), number of pigs weaned per sow (PS), and number of pigs weaned per sow per year (PSY) in the last six months. When the respondents left any response blank, it was considered a non-response and was not included in the analysis.

### 2.2. Detection of PPVs, PCV2, PCV3, PCV4, and PRRSV

Blood samples were centrifuged at 3500 rpm/10 min and stored at −80 °C until processing. Individual extraction was performed from 200 μL of each serum (*n* = 234) using the High Pure Viral Nucleic Acid kit^®^ (Roche, Mannheim, Germany) following the manufacturer’s recommendations. Each extraction was divided into two aliquots, one for detecting DNA viruses and another for detecting PRRSV; these were stored at −80 °C until processing. The detection of nPPVs (PPV2 through PPV7) was achieved via real-time PCR using the specific primers reported in the literature as follows: PPV2 [10], PPV3 [41], PPV4 and PPV5 [24], PPV6 [42], and PPV7 [43]. Amplification was detected with SYBR Green. To confirm DNA integrity, each sample was tested for the porcine β-actin gene using the primers reported in [44]. Appendix A contains the list of primers used and the size of the amplicons for the viruses studied. For the construction of positive controls and the standardization of real-time PCRs for each of the nPPVs, field samples (tissues) previously diagnosed (by conventional PCR) as positive were used. The amplicons were obtained using the primers mentioned above. These amplicons were cloned in the TOPO plasmid cloning kit. TA^®^ (Invitrogen, Carlsbad, CA, USA) was transformed into One Shot^®^ Chemically Competent *E. coli*. The presence of the insert and the correct direction was determined via sequencing by a commercial sequencing company, SSiGMol (Servicio de Secuenciación y Análisis Molecular, Instituto de Genética, Universidad Nacional de Colombia, Bogotá). The recombinant plasmids were purified using the plasmid maxi kit^®^ (Qiagen, Hilden, Germany) and quantified via OD260 on a Nano200^®^ spectrophotometer (Thermo Scientific, Wilmington, DE, USA). Plasmids were diluted in RNase-free H_2_O to obtain a stock concentration of 10^8^ copies of plasmid DNA per microliter. Standard dilutions were prepared from the stock by serial dilutions in Log10 in RNase-free H_2_O. From the construction of the standard curves, a Ct < 35 was established as the detection limit (declaration of a sample as positive) for each of the nPPVs (PPV2 through PPV7). The specificity of each real-time PCR (PPV2 through PPV7) assay was determined using the recombinant plasmids constructed for each nPPV (mentioned above). The assay specificity was also evaluated using a panel of several swine DNA pathogens, including PCV2, PCV3, PPV1 to PPV7, and *Mycoplasma hyopneumoniae*. Real-time PCR was performed using the primers at a concentration of 0.4 µM in 20 µL (SsoAdvanced Universal SYBR Green Supermix^®^-BIORAD). All samples were processed in duplicate on LightCycler 480^®^ equipment (Roche, Burgess Hill, UK). The amplification profile for PPV2 to PPV7 was a 10 min activation cycle at 95 °C followed by 45 cycles of 1 min at 95 °C, 24 s at 60 °C, and 2 s at 72 °C. A melting curve analysis was performed by monitoring the fluorescence of the SYBR-Green signal from 65 °C to 95 °C. The qPCR limit of detection (LoD), the coefficient of determination, and the efficiency were evaluated using a serial 10-fold dilution curve. The qPCR efficiency (E) was evaluated through the formula E = 10(−1/slope) − 1. The slope was calculated using linear regression between the crossing points (Cq) and corresponding log-transformed viral titers. R2 summarizes the goodness-of-regression line fit in explaining the relationship between dilution and Cq. The LoD was defined as the lowest viral amount detected in at least 50% of replicates for each dilution. Intra-laboratory repeatability was estimated by choosing three dilutions representing high, medium, and low viral concentrations and having them tested by two operators on two days. The selected DNA dilutions were 10^−1^, 10^−4^, and 10^−8^ [45]. Additionally, in order to confirm the sera declared nPPV-positive by real-time PCR, broader genomic regions were amplified through individual end-point PCR for each nPPV, performed using specific primers reported by [14] as follows: PPV2 (563 bp), PPV3 (514 bp), PPV4 (416 bp), PPV5 (959 bp), PPV6 (650 bp), and PPV7 (241 bp) (primers reported in [43]). The positive controls for this part were DNA extractions from tissues previously established as positive, which were part of the laboratory sample bank. These PCRs were carried out in a single reaction for each nPPV using 0.25 µL of Taq polymerase (5 U/mL) (Go Taq flexi-Promega^®^), 5× Taq buffer (2.5 µL), 1 µL of each primer (20 µmol/L), and 2 µL of extracted DNA from each serum. The thermal cycling conditions for each PCR were 94 °C for 5 min, followed by 35 cycles of 94 °C for 30 s, 58 °C for 30 s, 72 °C for 30 s, and a final elongation step at 72 °C for 10 min. The positivity of the nPPVs was also confirmed by sequencing the amplicons obtained in the end-point PCR. For this purpose, the amplicons were purified and sent to be sequenced with the primers already mentioned [14,43] in both directions at the commercial sequencing facility SSiGMol (Servicio de Secuenciación y Análisis Molecular), Instituto de Genética, Universidad Nacional de Colombia, Bogotá.

In the case of PRRSV, the extraction aliquot separated for this purpose was subjected to cDNA synthesis using the High-Capacity RNA-to-cDNA Kit^®^ (Thermo Scientific). Once the PRRSV-cDNA was obtained, real-time PCR was performed using specific primers and a Taqman probe, as reported by [46], as well as Taqman probes for PPV1 [47], PCV2 [48], and PCV3 [49]. The list of primers used and the size of the amplicons for these viruses can be found in Appendix A. This technique is standardized and routinely used in the laboratory to detect the viruses mentioned above, and the detection limit to declare a positive sample is a Ct of <37. Positive controls were cloned plasmids with specific sequences for each virus. Real-time PCR was performed using each probe at 0.25 µM and primers at a concentration of 0.4 µM in 20 µL of LightCycler 480 Probes Master^®^—Roche. All samples were processed in duplicate on LightCycler 480^®^ equipment (Roche, Burgess Hill, UK). The amplification profile was a 10 min activation cycle at 95 °C followed by 45 cycles of 15 s at 95 °C, 45 s at 60 °C, and 2 s at 72 °C.

Two other viruses (PCV4 and PPV8) were also evaluated, which, at least until the samples were obtained (December 2020), had yet to be detected in Colombia. For PCV4, conventional PCR was used using specific primers and the protocol reported in [50]. At the same time, PPV8 was searched through conventional nested PCR (nPCR) using specific primers and the protocol reported in [5]. For these two viruses, our laboratory does not have positive controls.

### 2.3. Statistical Analysis

PCR results were reported as binary data (positive/negative) for each virus tested. The crude association between PPVs and PCV2, PCV3, and PRRSV was estimated by univariate analysis (chi-square or Fisher’s) and considered significant when the *p*-value was <0.05. Then, the mean value for the farrowing rate (FR, x = 86.63), percent of mummies (MMs, x = 5.08), stillbirths (SBs, x = 5.37), and piglets per sow per year (PSY, x = 27.03) between herds was used to classify each herd as high or low for each RPP. Then, the association between high or low RPPs (FR, MM, SB, and PSY) and virus detection in gilts (positive or negative) for PCV2, PCV3, PPV1 through PPV7, and PRRSV was estimated using a univariate analysis (chi-square or Fisher’s) and considered significant when the *p*-value was <0.05. Then, the crude odds of detection of each significant association were estimated. To estimate the adjusted odds of reproductive performance (high vs. low) given the detection of each virus, we fitted a multivariable logistic regression model including all viruses for which the individual *p*-value was lower than 0.16, keeping in the final model only those significant (*p* < 0.05) or those that confounded the adjusted odds. Confounding was defined as a 10% difference between the crude and adjusted coefficients in the logistic regression model. Logistic regression methods were used based on [51] (chapters 3, 6, and 7). Finally, a multiple-correspondence analysis (MCA) was performed to describe the relationships between the independent categorical variables related to herd size, acclimatization practice, replacement rate, source of the gilts, and the presence or absence of each virus using the “FactoMineR” and “Factoextra” packages from the computational language for statistical analysis RStudio version 2022.12. [52,53].

## 3. Results

### 3.1. Herd Management and Reproductive Performance Parameters

Thirty-four surveys (85%) and thirty RPP records (75%) were obtained from the 40 herds sampled. Data collection was performed and the samples were collected between June and December 2020. The RPPs and management practices are summarized in Table 1. While the majority of herds (*n* = 16, 40%) were in the small category (100–300 sows), 30% (*n* = 12) had up to 1000 sows, and 13% (*n* = 5) had more than 1000 sows. Information regarding the sow inventory was unavailable for the remaining 17% (*n* = 7) of the herds.

### 3.2. Detection Rates of Porcine Parvovirus in Gilts

Novel PPV species (PPV2 through PPV7) were detected in Colombia during the present study (Figure 1A, Appendix A). Additionally, the samples were evaluated for PPV8, but all were negative. At the herd level (*n* = 40), the most prevalent PPV was PPV3 (67.5%, *n* = 27), followed by PPV5 (40%, *n* = 16), PPV6 (32.5%, *n* = 13), PPV2 (22.5%, *n* = 9), PPV1 (17.5%, *n* = 7), PPV4 (15%, *n* = 6), and PPV7 (5%, *n* = 2) (Figure 1B, Appendix A). At the individual level (*n* = 234 sera samples), PPV3 was also the most common PPV species found in gilts (40.1%, *n* = 94), followed by PPV5 (20.5%, *n* = 48), PPV6 (17%, *n* = 40), PPV2 (9.8%, *n* = 23), PPV4 (4.2%, *n* = 10), and PPV7 (1.3%, *n* = 3). When the PRF viruses (PPV1, PRRSV, PCV2, and PCV3) were examined in gilt sera samples, PRRSV was the most prevalent, with 63.6% testing positive (149/234), followed by PCV2 with 53.4% (125/234), PCV3 with 30.7% (72/234), and finally PPV1 with 14.5% (34/234) (Figure 1A, Appendix A). When the presence of PRF viral agents was analyzed at the herd level, the most frequent was PCV2 (85%), followed by PRRSV (82.5%) and PCV3 (57%) (Figure 1B, Appendix A). Additionally, we evaluated PCV4, but it was negative for all samples.

Figure 2 shows the percentage of presence of the viruses detected in each of the five regions studied in Colombia. PRF primary viruses (PPV1, PCV2, and PRRSV) were detected in all regions along with PPV3. The remaining nPPVs (PPV2, PPV4, PPV5, and PPV6) were distributed among four regions, while PPV7 was only present in two (Atlántico and Valle del Cauca).

### 3.3. Virus Detection at the Herd Level and Infection at the Individual Level

This study detected at least two or more viruses in each herd (Appendix A). Two viruses were detected in 7.5% (3/40) of the herds, three in 22.5% (9/40), four in 25% (10/40), five in 22.5% (9/40), and six or more viruses were found co-circulating in 20% of the herds (8/40). At the individual level (Appendix A), only 4.2% (10/234) of the gilts were negative for all viruses; a single infection was only found in 16.6% (39/234) of the gilts, and PRRSV (35.8%, 14/39), PPV3 (23%, 9/39), and PCV2 (17.9%, 7/39) were the most frequent viruses. Double infection was detected in 29.4% (69/234) of the gilts, and the most common coinfection was PCV2/PRRSV (30.43%, 21/69). Coinfection of any PPV species with PCV2 and with PRRSV was found in 20% (14/69) and 21.7% (15/69), respectively. Discriminating the above coinfections, the most frequent were PCV2/PPV6 (10.1%, 7/69), PRRSV/PPV3 (8.7%, 6/69), and PRRSV/PPV1 (8.7%, 6/69). Triple coinfections were found in 27.3% (64/234) of the gilts, while 15.3% (36/234) had quadruple coinfections, 5.1% had quintuple coinfections (12/234), and 1.7% had six or more viruses (4/234) simultaneously (Appendix A). The single and concurrent infections of PPV species (PPV1 through PPV6) with PCV2, PCV3, and/or PRRSV are illustrated in Figure 3. No coinfection was found in any serum involving more than five PPV species. Coinfections were found in all five regions studied and in all herds evaluated. 

The number of coinfections (double to octuple) and their diversity revealed extensive information, so we classified only dual coinfections where a PPV and some PRF viruses (PCV2, PCV3, and PRRSV) were present, independent of the number of viruses present in coinfections. Thus, the most frequent concurrences in gilts were PRRSV/PPV3 (27%), followed by PCV2/PPV3 (16%), PRRSV/PPV6 (15.8), and PRRSV/PPV5 (15.4%) (Table 2).

### 3.4. Association between PPVs (PPV1 through PPV7) and PCV2, PCV3, PCV4, or PRRSV

These analyses examine the association between PPVs (PPV1 through PPV7) in gilts and PCV2, PCV3, or PRRSV. A significantly higher number of PPV1 infections was found in PCV2-positive gilts versus PCV2-negative gilts (*p* < 0.01). At the same time, the number of PPV3 infections was higher in PCV2-negative gilts compared to PCV2-positive gilts (*p* < 0.01) (Figure 4A). In the case of PCV3, a higher number of PPV6 infections was observed in PCV3-positive gilts compared to PCV3-negative gilts (*p* < 0.01), but there was a higher number of PPV1 (*p* < 0.01) and PPV4 (*p* < 0.05) infections in PCV3-negative gilts compared to PCV3-positive gilts (Figure 4B). Furthermore, the number of PPV5 infections was higher and significant (*p* < 0.05) in PRRSV-positive gilts than in negatives. On the contrary and also significantly (*p* < 0.05), we found higher numbers of PPV3 infections in PRRSV-negative than in PRRSV-positive gilts (Figure 4C). Although not statistically significant, a higher number of PPV3 infections was also found in PCV3-positive gilts (*p* = 0.12), and PPV6 infections were higher in PRRSV-positive gilts (*p* =0.07).

### 3.5. Association between PPVs, PCV2, PCV3, and PRRSV Status in Gilts and Performance Parameters in Sows 

The crude association between virus detection (positive or negative) for PPVs, PCV2, PCV3, PRRSV in gilts, and high or low reproductive performance parameters (RPPs) [(above or below sample average farrowing rate (FR), percentage of mummies (MMs), percentage of stillbirths (SBs), and number of pigs weaned per sow per year (PSY)] was only statistically significant for PCV3 and FR (*p* = 0.0043), indicating that the crude odds of a low farrowing rate [below sample average (86%)] were 94% lower in herds with PCV3-positive gilts than in herds with PCV3-negative gilts (OR 0.06, 95% CI: 0.007, 0.36). Furthermore, a low FR (below sample average 86.6%) tended to be associated with the detection of PPV4 (OR: 4.3, *p*-value = 0.16) and PPV5 (OR: 4.0, *p*-value = 0.13) in gilts. However, the multivariable logistic regression model (full model) used to estimate the adjusted association between the FR (lower vs. higher than average) and PCV3, PPV4, and PPV5 indicated that the effect of PPV5 was not significant (*p*-value = 0.11) and therefore it was dropped from the model. The reduced model indicated that the odds of a low farrowing rate [lower than average (86%)] were 96% lower in herds with PCV3-positive gilts than in herds with PCV3-negative gilts (OR: 0.04, 95% CI: 0.002, 0.30) after adjusting for PPV4 status (Table 3). Interestingly, the odds of a low FR were 9.6 times higher in herds with PPV4-positive gilts than in herds with PPV4-negative gilts after adjusting for PCV3 status. The difference in the coefficient estimates for PCV3 in the crude (−2.76) versus the adjusted (−3.18) model indicated that PPV4 confounds the association between PCV3 and FR (coefficient estimates changed more than 10%). Although not statistically significant, it was also found that a high percentage of MMs (above average) tended to be associated with herds in which gilts tested positive for PCV3 (*p* = 0.07), PPV5 (*p* = 0.13), and PRRSV (*p* = 0.14); a high SB percentage was associated with positive PPV1 gilts (*p* = 0.07); and a low PSY number was associated with PPV3-positive gilts (*p* = 0.12). All other associated measures were considered not relevant (*p* > 0.20).

### 3.6. Multiple Correspondence Analysis—MCA

An MCA was carried out to estimate the relationships between the management practices of gilts (Table 1) and the presence of viruses in serum samples from gilts. Figure 5 gives the MCA solution for these variables. Dimension 1 was influenced primarily by a large herd size (>1000 breeding females), the replacement rate (>50%), acclimatization practices, and herds with the presence of more than six viruses. Dimension 2 was mainly influenced by a medium herd size (100–500 breeding females), the replacement rate (<50%), no acclimatization practices, and herds with the presence of less than six viruses. PPV4 and PPV7 were present in herds with the simultaneous presence of seven or eight viruses and with more than 1000 sows with a replacement greater than 50%.

## 4. Discussion

Novel PPVs (PPV2 to PPV7) are infecting swine populations [9,15,52], and the results of some studies suggest that some of them could be associated with other viruses (PPV1, PCV2, PCV3, and PRRSV) in the presentation of the PRF [15,54]. However, substantial evidence for this proposal still needs to be made available. Additionally, the effect of virus interactions on swine health and production under field conditions is unknown. This study demonstrated that multiple nPPVs could coexist in both the population and individual gilts. Likewise, these nPPVs were detected in coinfections with other PRF viruses (PPV1, PCV2, PCV3, or PRRSV) and, even more importantly, may be associated with the RPPs of sows such as the farrowing rate. Understanding the synergic effect of these endemic and epidemic coinfections is essential to designing measures to prevent infection, minimize transmission, and reduce their negative effect on swine health and production. Since the emergence of nPPV species, they have been reported worldwide in different countries, swine populations, and sample types [9,13,28]. The detection rate of PPV differs between studies and varies significantly depending on the population evaluated (domestic pigs or wild boar), sample type, and diagnostic techniques [12,23,34]. In most population-based studies where nPPVs have been detected, samples such as sera, stools, and oral fluids have been evaluated, and the population groups evaluated were mainly nursery and grow-finish pigs [9,27]. In the present study, PCR was performed on serum samples to look for viremia for nPPVs; it was assumed that this type of sample would be significant for viral detection. However, it is necessary to remember that the tropism of nPPVs is unknown, and the best sample type has not been determined. Therefore, performing PCR on other types of samples could change the result. The lack of knowledge about the biological behavior of nPPVs leads to the limitation of the best sample for detection by PCR.

The information available in gilts on the prevalence and the role of viral co-detection (at the population level) or coinfection (at the individual level) is scarce and should be further investigated. In this sense, viral coinfections could influence the disease pattern and modulate the immune response in a mono-infection [55]. In the present study, the factors that could favor coinfections were (i) PCV2 and PRRSV, which were the viruses most detected in coinfections with nPPVs, and several studies have shown that these two viruses can alter the host’s immune response (they are immunosuppressive) [56,57,58]. Specifically, the mechanisms (type of immunity) involved in promoting coinfections with nPPVs are unknown. (ii) The immune systems of the gilts (study population), which were not sufficiently challenged to subvert the infection compared to multiparous sows [59]. (iii) Biosecurity measures in the herds.

This study evaluated individual sera from gilts in 40 swine herds in Colombia. We found that 1.8–40% of samples were positive for nPPV species (PPV2 through PPV7), highlighting their wide distribution among the swine population within five regions of Colombia. Our results provided new insights regarding co-detection and coinfection between nPPVs and other viruses involved in PRF, such as PCV2, PCV3, and PRRSV. Research regarding coinfections between PCV2, PRRSV, PPV1, PCV3, and nPPVs in breeding sows is limited. However, two previous studies helped detect PPVs [27,60]. In the first, PPV1 to PPV4 was evaluated in adult pigs (older than 1 year old,) finding 17.5% as PPV3-positive [60]. In the second study, PPV1 through PPV7 were examined in serum pools from breeding sows as a whole (sows and gilts), and it was found that PPV3 and PPV6 were the most prevalent (10.5%) [27]. Our study coincides with these two previous reports in that PPV3 was the most prevalent PPV found in gilts, although our study found a higher percentage of PPV3-positive samples (40%); this higher percentage could be associated with the specific population targeted (gilts from 180–210 days of age) vs. sows [27] or growing pigs [60], highlighting the role that gilts may play in the epidemiology of PPV3. Other studies have shown that gilt populations in sow herds play a vital role in transmitting other viral infections, such as SIV [61], and they are essential in the persistence of PRRSV in endemic populations [36]. Future studies are needed to better understand the epidemiology of PPV3 in swine populations, particularly the role of gilts in viral transmission and maintenance. Likewise, it is necessary to determine the effect of the differences in prevalence found in different regions of the world for nPPVs and its impact on pig health and production.

There are three previous studies, one in Mexico [15], another in South Korea [62], and another in Italy [54], where the authors evaluated the nPPVs associated with PRF (specifically in aborted fetuses), which found the presence of these viruses (PPV2 through PPV6) with varied prevalences. These studies agree that the lowest prevalent was for PPV4. Likewise, the prevalence of PPV5 and PPV6 ranged between 18% and 56%, respectively; PPV3 was found in different proportions (3% to 38%); and PPV2 was the most variable (2% to 95%). Of these three studies, only the one from Italy [54] evaluated PPV7, finding a prevalence of 14.5%. Regarding this latter virus, most existing reports address its detection without looking for other nPPVs; in most cases, it has been found in aborted fetuses. When comparing our results with the other three reports, we also found that PPV4 had a low prevalence, and the amount of PPV5 and PPV6 detected was similar to these reports. As previously noted, in our study, the most prevalent was PPV3, similar to studies where sow sera were evaluated [27,60]. Additionally, we found that PPV7 was the least prevalent (1%); this difference with [54] and with studies that evaluated only PPV7 could be associated with the type of sample (in our study, sera) since the highest prevalences were reported in aborted fetuses (up to 50%) [29]. It can be established, from our own and others’ results, that nPPVs could be participating in PRF outbreaks, and particularly with our results, this proposal is complemented by drawing attention to the presence of nPPVs in the population of gilts that would begin their reproductive stage infected with these viruses. Several previous studies have demonstrated the impact caused by PRF primary viruses (PRRSV [63], PCV2 [64], and PPV1 [65]) on the reproductive performance in gilts and sows, including PCV3, which has been detected in sows with PRF [38,66]. Investigations mainly determine the impact of a single virus and not of coinfections. The three studies mentioned above [15,54,62] also evaluated the association of nPPVs with PRF viruses (PCV2, PCV3, and PRRSV). In the studies from Mexico and South Korea, dual coinfections between PCV2 and nPPVs (PPV2 through PPV6) were evaluated, and all of them were found, the most prevalent being PCV2/PPV2, PCV2/PPV5, and PCV2/PPV6. Additionally, they looked for correlations between the detected viruses. The Mexican study found significant correlations between PCV2-positive samples and PPV5 and PPV6, unlike South Korea, where they found no correlation. The Italian study evaluated the association of PCV2, PCV3, and PRRSV with nPPVs, finding coinfections of PCV2/PPV3 to PPV7; PCV3/PPV3 to PPV7, and PPRSV/PPV2 to PPV7; the most frequent being PCV2/PPV3, PCV3/PPV3 and PPV6, and PRRSV/PPV2 and PPV3. In this case, they also found no correlation between coinfections and PRF. Our study revealed coinfections of PCV2/PPV2 to PPV7; PRRSV/PPV2 to PPV7; and PCV2/PPV2, PPV3, PPV5, PPV6, and PPV7. The most frequent were PCV2/PPV3; PCV3/PPV3; PCV3/PPV6; PRRSV/PPV3; and PRRSV/PPV5. It must be noted that we found correlations between a high amount of PPV6 in PCV3-positive gilts and PPV5 in PRRSV-positive gilts. Additionally, there was a high amount of PPV3 in PCV2-negative gilts, PPV4 in PCV3-negative gilts, and PPV3 in PRRSV-negative gilts. These associations constitute the first report in the literature and the first step to establishing if there is a synergistic effect of these coinfections, which, in the case of PRRSVs, could be closely associated with their immunomodulatory effect [58]. These particularities should be studied further. Other studies have shown an association between PCV3 and PPV7 [67] that we did not find.

Moving on to PPV1, its genomic detection has been reported in most studies despite the vaccine’s routine use. In general, its prevalence is less than 30%, although high prevalences of PPV1 (44%) have been reported in boars in Italy [34]. It is striking that the prevalence of some nPPVs (PPV2, PPV3, PPV6) is, on average, higher than PPV1. This virus has also been found in coinfections: PPV1/PCV2 (less than 10%), PPV1/PCV3 (25%), and PPV1/PRRSV (9%). Our findings are similar to these averages, at PPV1/PCV2 (7.5%), PPV1/PCV3 (4.3%), and PPV1/PRRSV (8.5%). We also found an association with a higher amount of PPV1 in PCV2-positive gilts, as has already been reported in other studies in sera, but in pigs at different stages of growth and associated with PCVADs [68,69]. In our case, the synergy reported between PPV1 and PCV2 in gilts could be significant in developing PRF when integrated into the herd’s reproductive cycle. Due to the general biological characteristics, controlling PPVs is difficult because they are resistant to adverse environmental conditions, survive for long periods, and resist inactivation by many disinfectants, depending on the species of PPV [70]. On the other hand, vaccination and gilt acclimation [exposure to feedback (manure, fetuses, and other tissues)] have been the most effective tools for controlling PPV1 [71]. However, the effect of feedback on the epidemiology of nPPVs is unknown and needs to be studied. Additionally, new PPV1 strains have emerged in recent decades with capsid protein (VP2) mutations, which have modified antigenic properties and a decreased cross-reactivity, conferred by the available commercial vaccines [72]. The nucleotide identity between PPV1 and nPPVs is approximately 40%, suggesting there is little or no cross-reactivity. 

No published works have estimated the effect of nPPVs on PRF and, therefore, the consequences on the RPPs. This study demonstrated several exciting findings. First, PCV3 was associated with the FR (*p* < 0.05), and PPV4 and PPV5 tend to be associated with the same performance parameter. Additionally, we demonstrated that the association between PCV3 and FR is different (more than 10% difference) when PPV4 is considered, indicating that PPV4 confounds the association between PCV3 and the FR. These results suggest that PPV4 could be a virus involved in PRF, as has already been proposed with PPV1 and PPV7 [67]. The results of this work are the first to establish, in a field situation, the confounding effect of nPPVs and PRF. However, more studies are needed to determine the direct causality of PPVs in mono- and coinfections, and whether PPRs, at the herd level, can be affected by other pathogens and other factors [73]. 

Another significant contribution of the present study is that although the risk factors associated with the presence of the nPPVs were not established, the MCA determined two things: first, in the herds where PPV4 or PPV7 was present, there was a greater co-circulation of viruses (up to six), and second, this greater co-circulation also occurred in herds with a higher number of sows (≥1000 breeding females) and replacement rates of ≥50%, highlighting the impact that population size and replacement rates may have in the epidemiology of swine viruses, as was previously described by other authors [74,75]. These conditions may favor the establishment of nPPVs in swine populations due to PPVs’ structural and physicochemical properties [71]. It is important to highlight certain particularities within the nPPV species. PPV4, for example, has a circular DNA genome, unlike other PPVs with linear DNA. This characteristic may favor its persistence in pigs [21] because it can remain in episomal form in the nucleus of infected cells for long periods and may be more resistant to nucleases. Moreover, PPV7 belongs to a different viral subfamily than the other parvoviruses identified to date, which gives it unique characteristics compared to other members of the *Parvoviridae* family [76].

Finally, the results presented here can add to the two hypotheses about nPPVs. The first proposes that most viruses can cause asymptomatic infections despite being very well associated with a disease (the iceberg concept of viral infection); they are commensals and are part of the pig virome [25,77]. The second proposes that they cause persistent infection [78]. Our results showed that nPPVs were present in the serum of healthy gilts; this could initially indicate that they could be present in these gilts without being involved in disease, supporting the first hypothesis. However, PRF viral agents were also detected in healthy gilts. The latter observation leads to the proposal that for both the primary viruses and the nPPVs, there must be a threshold (probably the viral load) for them to act as direct (putative) pathogens (causing lesions) or indirect pathogens (causing immunosuppression or immunomodulation). The second hypothesis states that some nPPVs could be putative agents of disease and act in PRF [15] or the PRDC [14,79]. In the present study, we found an association between PPV4 and a lower RR and determined an association with some RPPs; likewise, we identified PPV4 in coinfections with other PRF viruses. These results could open the way to consider, in this case, PPV4 as an active agent in PRF, supporting the second hypothesis [68,80]. In order to support these hypotheses, it is necessary to design studies aimed at isolating these viruses and precisely establishing their pathogenesis along with their epidemiological behavior, thus coming closer to establishing whether or not they impact pig health. In light of our results, it is clear that the gilts studied were infected with several viruses, such as nPPVs, at insemination. Therefore, our scientific recommendation is to follow up via diagnosis of primary PRF viruses (PPV1, PCV2, and PRRSV) in gilts and multiparous sows to better understand their overall behavior during the reproductive cycle. From the producers’ perspective, a permanent diagnosis of these viruses is also recommended, along with a review of the control plans (vaccination and feedback exposure) against primary PRF viruses.

## Figures and Tables

**Figure 1 vetsci-11-00185-f001:**
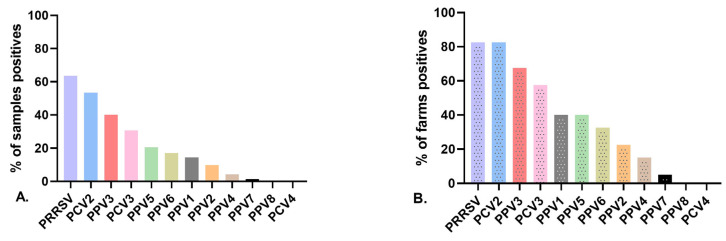
Prevalence of PPVs (PPV1 through PPV8), PCV2 to PCV4, and PRRSV in gilts from 40 herds in the five regions with the highest swine production in Colombia. (**A**) Sample prevalence and (**B**) herd prevalence.

**Figure 2 vetsci-11-00185-f002:**
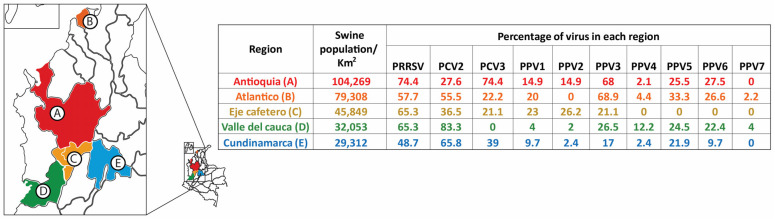
Map of Colombia indicating the five geographical regions evaluated (segregated by pig population density) corresponding to those with the highest pig production in Colombia: A. Antioquia; B. Atlantico; C. Eje Cafetero; D. Valle del Cauca; and E. Cundinamarca. The detection percentage obtained for each virus in each region is shown. The map was created using the National Department of Statistics of Colombia (DANE) database (https://geoportal.dane.gov.co/acerca-del-geoportal/acerca/#gsc.tab=0) (accessed on 12 January 2024) and adapted using the QGIS software version 3.34.5 available online: https://qgis.org/es/site/ (accessed on 12 January 2024).

**Figure 3 vetsci-11-00185-f003:**
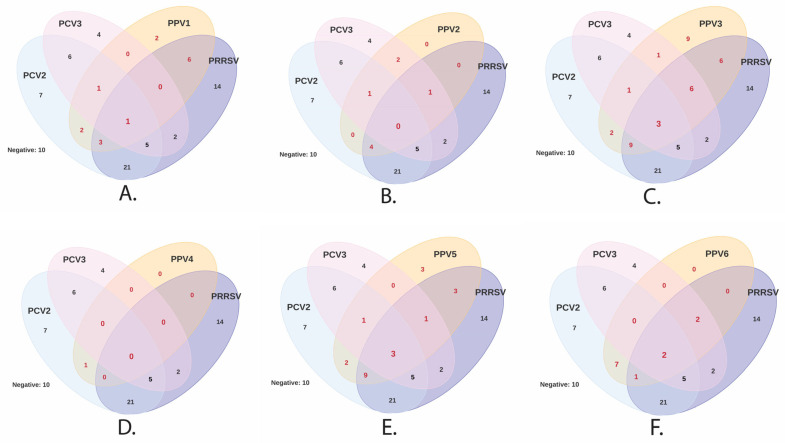
Detection of any PPV species (PPV1 through PPV7) and other swine viruses (PCV2, PCV3, PRRSV) in gilts from 40 herds in Colombia. Venn diagrams illustrate the number of samples with single infections (areas of a single color) and coinfections (overlapping bubbles) established between PPVs [PPV1 (**A**), PPV2 (**B**), PPV3 (**C**), PPV4 (**D**), PPV5 (**E**) and PPV6 (**F**)] and other viral agents (PCV2, PCV3, and PRRSV) involved in porcine reproductive failure. PPV7 was not illustrated because of the low detection during the study.

**Figure 4 vetsci-11-00185-f004:**
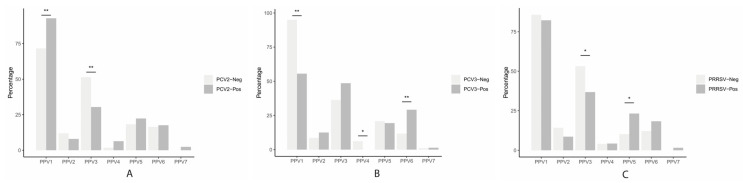
Association between PPV (PPV1 through PPV7) detection and PCV2 (**A**), PCV3 (**B**), and PRRSV (**C**) in gilts from 40 herds in Colombia. The bars illustrate the percent (y-axis) of positive or negative samples for PCV2, PCV3, and PRRSV. The x-axis represents each PPV evaluated. Significant (*) or highly (**) significant differences are indicated.

**Figure 5 vetsci-11-00185-f005:**
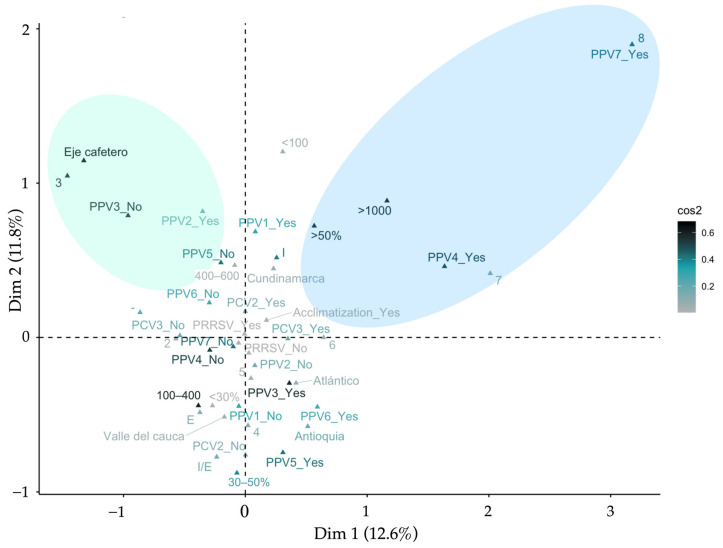
Gilt management practices and the presence of PPVs (PPV1 through 7), PCV2, PCV3, and PRRSV in Colombian swine farms: object scores of a two-dimensional multiple-correspondence analysis solution and two-step cluster solution [40 farms used in the analysis; dimension 1 (x-axis); dimension 2 (y-axis)].

**Table 1 vetsci-11-00185-t001:** Summary of herd management and reproductive performance parameters (RPPs).

Variable	Variable Categories	Frequency
Herd size	Small (100 to 300 breeding females)	16 (40%)
	Medium (301 to 1000 breeding females)	12 (30%)
	Large (≥1000 breeding females)	5 (12.5%)
	No response *	7 (17.5%)
Acclimatization practice	Yes	9 (22.5%)
	No	25 (62.5%)
	No response *	6 (15%)
Replacement rate	≥50%	9 (22.5%)
	30–50	16 (40%)
	≤30	2 (5%)
	No response *	13 (32.5%)
Source of the gilts	External (from other farms)	8 (20%)
	Internal (from the same farm)	19 (47.5%)
	Internal and external	7 (17.5%)
	No response *	6 (15%)
Farrowing rate (mean 86.6%)	Herds with a mean <86.6	11 (27.5)
	Herds with a mean >86.6	18 (45%)
	No response *	11 (27.5%)
Percentage of stillbirths (mean 5.4%)	Herds with a mean >5.4	15 (37.5%)
	Herds with a mean <5.4	16 (40%)
	No response *	9 (22.5%)
Percentage of mummies (mean 5.1%)	Herds with a mean >5.1	10 (25%)
	Herds with a mean <5.1	21 (52.5%)
	No response *	9 (22.5%)
Number of pigs weaned per sow (mean 11.2)	Herds with a mean >11.2	17 (42.5%)
	Herds with a mean <11.2	13 (32.5%)
	No response *	10 (25%)
Number of pigs weaned per year (mean 27)	Herds with a mean >27	15 (37.5%)
	Herds with a mean <27	9 (22.5%)
	No response *	16 (40%)

* no response: respondents left these answers blank, and hence they were not included in the analysis.

**Table 2 vetsci-11-00185-t002:** Frequency of porcine parvoviruses (PPV1 through PPV7) found in the gilts (40 herds, *n* = 234 sera) in double coinfections with PCV2, PCV3, and PRRSV.

PPVs	PCV2 *n* * (%)	PCV3 *n* (%)	PRRSV *n* (%)
PPV1	18 (7.7)	10 (4.3)	20 (8.5)
PPV2	10 (4.3)	9 (3.9)	15 (6.4)
PPV3	38 (16.2)	35 (15)	64 (27.3)
PPV4	8 (3.4)	0 (0.0)	7 (3)
PPV5	28 (12)	14 (6)	36 (15.4)
PPV6	22 (9.4)	21 (9)	37 (15.8)
PPV7	3 (1.3)	1 (0.4)	3 (1.3)

* Number of positive sera of 234 evaluated.

**Table 3 vetsci-11-00185-t003:** Multivariable logistic regression model between FR (below vs. above average) and PCV3 and PPV4 status of gilts (positive and negative).

Variable	Regression Coefficient	*p* Value	OR *	95% CI
Intercept	0.4813	0.4487		
Positive PCV3	−3.1804	0.0071	0.04	0.002–0.30
Positive PPV4	2.2666	0.0993	9.64	0.84–266.18

* Adjusted ORs for farms with a farrowing rate below the average and positive virus detection versus farms with a farrowing rate above the average and negative virus detection.

## Data Availability

The original contributions presented in the study are included in the article/Appendix A. Further inquiries can be directed to the corresponding author.

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
