# Peer review of "Infection and Coinfection of Porcine-Selected Viruses (PPV1 to PPV8, PCV2 to PCV4, and PRRSV) in Gilts and Their Associations with Reproductive Performance"

_vetsci, 2024, doi:10.3390/vetsci11050185_

Round 1
Reviewer 1 Report
Comments and Suggestions for Authors
This study explored the presence of seven new porcine parvoviruses (PPV2 to PPV7) in Colombian gilts, along with their co-infections with PPV1, PCV2, PCV3, and PRRSV, to assess any links to sow reproductive performance parameters (RPP). Sampling 234 healthy gilts across 40 herds in five regions, real-time PCR detection revealed prevalent PPV3 (40%), PPV5 (20%), and PPV6 (17%), with no findings of PCV4 or PPV8. Coinfections varied from dual to sextuple, notably between PCV3 and PPV6, and PRRSV with PPV5.
Before publishing, some points need to be addressed:
1. In Table 1, which summarizes herd management and reproductive performance parameters (RPP), the category 'no response' requires clarification. Could you please explain the meaning of this category either in the table caption or within the main text for better reader comprehension?
2. The manuscript suggests a high likelihood of coinfection among the swine populations studied. Could you elaborate on the underlying factors contributing to this increased risk of coinfection?
3. The resolution of the figures, particularly Figure 4 and Figure 5, is notably low, making the text within these figures difficult to see. Enhancing the quality of these figures is essential for readers to accurately interpret the data presented.
4. Several abbreviations are used throughout the text without being defined, such as FR, MM, SB, and PSY mentioned in line 308. For the sake of clarity and to accommodate readers unfamiliar with these terms, please define all abbreviations at their first occurrence.
5. The manuscript details the use of specific diagnostic techniques to detect novel Porcine Parvoviruses (nPPVs). Could you provide information on how these methods were validated in terms of their sensitivity and specificity?
6. How do the findings on the prevalence of nPPVs and their co-infections compare to existing literature, particularly in different geographical regions or farm management systems?
7. How might the findings of this study influence current strategies for the prevention, control, and management of nPPVs and associated reproductive failures in swine populations?

Good
Author Response
We want to thank reviewer 1 for his valuable contributions. The answers to your questions and suggestions are attached in the attached file.

Reviewer 2 Report
Comments and Suggestions for Authors
The article written by Vargas-Bermúdez et al. describes the detection of novel parvoviruses in Colombia in sows sampled from different farms. The detection occurred on serum samples, and coinfection with other swine pathogens was evaluated. The work is quite well written, although there are some necessary changes in the abstract and discussion sections. The work presents methodological rigor, although there are aspects that need to be integrated, improved, or justified. In my opinion, the article, after careful review, can be considered for publication. Below are some of my specific comments.
Simple summary:
Line 12: The sentence “Pig production is based on the reproductive performance of the sows” should be better articulated.
Line 13: Why are some pathogens considered traditional?
Line 20: “(increase)”? Please delete it.
Abstract:
Line 26: “nPPVs” Don't use acronyms that are only clarified in later sections. Also line 35, “(RPP)”
Line 31: Is this difference statistically significant? The authors could add this information.
Material and Methods:
Line 103: The number of sows sampled (234) comes from which formula?
Line 106: The collection of samples by region is unclear.
Line 113: How do the authors estimate the sensitivity and specificity of a non-commercial test based on SYBR green PCR? Please remove these references.
Line 130: Why didn't you consider an internal control for the quality of the matrix and extraction (such as B-actin or GAPDH, etc.)?
Line 182: So how was the actual positivity confirmed? Because the authors did not consider sequencing some amplicons (in general for all screened pathogens),.
Lines 198–200: The multivariate analysis seems wrong to me in general; however, in materials and methods, it should be more in-depth.
General comment: Could the type of matrix used have influenced the results? This is an important limitation of the research that needs to be discussed.
Figures: some figures are poorly readable as they are of low quality. Please replace them with better-quality images.
General comment: The multivariate analysis may be wrong (just as the MCA analysis may be unnecessary). The ORs have excessively wide confidence intervals (such as 0.84–266.18).
Discussion:
Line 352: The sentence “Novel PPVs (PPV2 through PPV7) infect the swine population [9, 15, 50] and may be associated with PRF” could be better explained.
Line 354: What do the authors intend for “primary viruses”?
General comment: I recommend a recently published article in which coinfection between PPV and PCV-2 and 3 was found in the Campania region of Italy, in the reproductive organs of wild boars. This information could also be used in the introduction.
Comments on the Quality of English LanguageThe article is written clearly and precisely, except for a few sentences that could be rewritten.
Author Response
We want to thank reviewer 2 for his valuable contributions. The answers to your questions and suggestions are attached in the attached file.

Reviewer 3 Report
Comments and Suggestions for Authors
The manuscript titled " Porcine parvoviruses (PPV1 through PPV8), PCV2 to PCV4, and PRRSV in gilts, their coinfections, and associations with the reproductive performance in sows" by Diana S. Vargas-Bermúdez et. al. to evaluate the effect of new virus on gilt herd and its effect on reproductive performance. It's a very interesting thing to do when there's no vaccine for a new virus. Below are certain points that need further attention.
1. The Figure 4 is not clear, suggest the author to modify it.
2. The Figure 5 is not clear, suggest the author to modify it.
3. The serum of 234 sows was collected for breeding and production. The authors have indicated the basic immunity of these sows, as well as the virus infection, which is more convincing, after all, the sow farm is basically immunized against PCV2 and PRRSV.
4. Serum was used to detect PRRSV and PCV2 nucleic acids. Did the authors consider using ELISA to detect antigens
Author Response
We want to thank reviewer 3 for his valuable contributions. The answers to your questions and suggestions are attached in the attached file.

Round 2
Reviewer 2 Report
Comments and Suggestions for Authors
The authors have addressed all my comments. I have just two further comments:
1) Title: The new title is long and not intuitive. I suggest changing the name of viruses to "selected viruses" or "selected pathogens.".
2) Simple summary: Please rephrase the sentence in line 22 (this is an observational study with limited information; only an experimental study could reveal the pathogenencity and dynamics of infection).
Comments on the Quality of English LanguageEnglish is ok
Author Response
We thank reviewer 2 for his comments and suggestions. The answers are attached.
